# Stratospheric Aerosol Climatology over Ethiopia and Retrieval of its **Size Distribution**

Milkessa Gebeyehu Homa<sup>\*1</sup>, Gizaw Mengistu Tsidu<sup>2</sup>, and Derese Tekestebrihan Nega<sup>1</sup>

<sup>1</sup>Jimma University, College of Natural Sciences, Department of Physics, Jimma, Ethiopia <sup>2</sup>Addis Ababa University, College of Natural Sciences, Department of Physics, Addis Ababa, Ethiopia

Correspondence to: Milkessa Gebeyehu Homa (mikemen.homa@gmail.com)

Abstract. Stratospheric aerosols play significant role both positively and negatively in Earth's energy balance and climate change. Its main sources are particulate matters which arises from either of natural or anthropogenic activities. In the context of our country, Ethiopia, the stratospheric aerosol climatology has not been studied yet. However, Ethiopia is undergoing a boom of infrastructural development like increase of urbanization, which comes with a boom of development like building

- and road constructions, expansion of industries, traffic density, etc, which contributes to air pollution and influences the solar 5 radiation budget of the earth-atmosphere system, which in turn influences the climate on the surface of the Earth by different ways. Hence, this study aimed to provide the stratospheric aerosol climatology for nearly 21 years extending from Oct., 1984 to Sept., 2005. The study was carried out by defining the stratospheric region from the temperature profile of the study area provided by Stratospheric Aerosols and Gas Experiment II (SAGEII) instrument aboard on Earth's Radiation Budget Satellite
- (ERBS). Then, the data was filtered out over Ethiopian region at four aerosol channels and the optical depth is used as input to 10 the Mie algorithm for aerosol size distribution (ASD) retrieval. Finally, it was observed that the spectral and vertical variation of the extinction is maximum between 17 - 25km and the total column aerosol optical depth (AOD) temporal variation shows nearly steadily increasing trend with maximum variation during spring. Furthermore, from the ASD result it was observed that the maximum size distribution was in April. This paves a clue about their sources to be mechanical process on the ground and
- 15 gas to particle conversion in the stratosphere with the dominant size distribution in the range of  $0.452 - 0.525 \mu m$  radius.

# 1 Introduction

Atmospheric aerosols are liquid or solid particulate matters suspended in the air. They are highly populated in the lower atmosphere. However, the mid-atmosphere aerosols play significant roles in the atmospheric sciences because of their contribution to the Earth's climate. Non-absorbing aerosols increase the albedo of the atmosphere and reduce the amount of solar radiation

20

(short wave) reaching the surface, but if it is absorbing aerosols in the shortwave (SW) range of the spectrum, energy is directly transferred to the atmosphere, and the effect is heating of the atmosphere and cooling of the underlying surface (Tzanis C. and Varotsos C.A., 2008).

The distribution of aerosols decreases with altitude and has a significant impact on the radiative balance and chemistry of troposphere (Chowdhary J. and Cairs B., 2002). The sources of these particulate matters could be from anthropogenic or

natural activities at the surface or within the atmosphere. The presence of these particulate matters in the Earth's atmosphere has significant impact (both positively and negatively) either directly or indirectly on human activities in particular and life on Earth in general. In addition, aerosols scatter and absorb shortwave (solar) and long wave (thermal infrared) radiation, thereby perturbing the energy budget of the Earth-atmosphere system and exerting a direct radiative forcing (RF) which is a measure of the net reduction or increase in the amount of surface-reaching or outgoing radiation due to total columnar aerosol burden

5

(CCSP., 2006), (Fu Q., et al., 1999).

Though this crude fact is there, in the context of our country, Ethiopia, the aerosol size distribution has not been studied yet for such long period. As Ethiopia is undergoing a boom of infrastructural development like increase of urbanization, which comes with a boom of development like building and road constructions, expansion of industries, traffic density, etc, which

- 10 contributes to air pollution by different ways. One of this is solid material loading to the atmosphere, mostly of dusts and minerals. The accumulation of these particulate matters affects the country's solar energy budget (solar radiation reaching the surface of the Earth), health, visibility (on aviation industry), etc. Hence, it is the right time to give the right attention to air quality, holistic ground based solar energy potential prediction and climate change impacts as it is a night mare issues nowadays.
- 15 Therefore, this work is the first to provide a long term data analysis report on aerosol climatology in Ethiopia. Its main goal is to investigate vertical and seasonal distribution and annual mean values of aerosol optical depth over Ethiopia using the stratospheric aerosols and gas experiments (SAGE) II instrument boarded on Earth's radiation budget satellite (ERBS) extinction data for a period of approximately 21 years extending from Oct., 1984 to Sept., 2005. The vertical distribution of aerosols at different channels of SAGE II and spatial distribution are also discussed. The trend of column amount aerosol optical depth (AOD) and the monthly mean size distribution function of aerosols for some of the selected years are retrieved
- 20 optical depth (AOD) and the monthly mean size distribution function of aerosols for some of the selected years are retrieved using Mie algorithm (King M.D. et al., 1980).

### 1.1 Sources and Removal of Atmospheric Aerosols

It is usual practice to group aerosols by their origins: those produced as a result of naturally occurring geological processes; those produced as a result of human activity ('anthropogenic'); and those produced by biological processes ('biogenic'). Dis-

- 25 tinction is also often made between tropospheric and stratospheric aerosols, as those often share different sources and have different radiative effects (Sayer A.M., 2006). After introducing the continuous atmospheric aerosol size distribution, Jung (1963) mentioned in (Hobbs P.V., 1993), also classified them geographically into maritime, continental, and background aerosols and by size into Aitken  $(0.001\mu m - 0.1\mu m)$ , large  $(0.1\mu m - 1\mu m)$  and giant  $(> 1\mu m)$  radius particles. Whitby (1973) in (Hobbs P.V., 1993), introduced the terms nucleation mode  $(0.001\mu m - 0.1\mu m)$ , accumulation mode  $(0.1\mu - 1\mu)m$
- and coarse particle mode (> 1µ)m. The nucleation mode was produced by gas-to-particle conversion (GPC), the accumulation mode by coagulation and heterogeneous condensation, and the coarse mode by mechanical processes. Volcanic sources (like: Mt.Pinatubo eruption in 1991 and Mount St. Helens eruption in July 22, 1980) are globally significant in their influence on the stratosphere .

5

There exists close relationship between atmospheric aerosols and the process of cloud formation and precipitation, implying a water cycle in the atmosphere would not be possible in the absence of aerosols. On the other hand, it is precisely this water cycle, made possible by aerosols that represents the most effective process for their removal from the atmosphere (Israel H. and Israel G.W., 1974). Some of the processes of removal of aerosols from the atmosphere include: Wet removal, dry removal, cloud formation, etc. The overall stratospheric aerosol cycle is shown in Fig.1 and discussed in (Thomason L. et al., 2007).

Figure 1. Schematics of overall stratospheric aerosol lifecycle (Thomason L.W. et al., 2007)

### 2 Methods

In this study the secondary extinction data from SAGE II instrument aboard the Earth Radiation Budget Satellite (ERBS) which was launched in October 5, 1984 was used. The instrument was able to provide high quality measurements of ozone, nitrogen dioxide, water vapor, and profiles of aerosol extinction at wavelengths centered at 386,452,525, and 1020 nano meters from

SAGE II instrument views a small portion of the sun through the Earth's atmosphere during the spacecraft's sunrise and sunset (using limb occultation technique). Data are obtained from the attenuation of the sun light due to scattering and absorption

15 by different atmospheric species. The spacecraft's motion just before entering or just after leaving the Earth's shadow provides vertical scanning through the atmosphere. Measurements taken from a tangent-height of 150 km, where there is no attenuation, provides a self-calibration feature for every event (Thomason L.W. et al., 2007).

<sup>10</sup> the mid-troposphere to as high as the lower mesosphere. All profiles are at 0.5km vertical resolution. These products are nearly global in coverage, with data spanning from  $80^0N$  to  $80^0S$ . The temporal coverage of SAGE II was from 5, Oct. 1984 to 8, Sep. 2005.

point the tropopause level over tropical region in general as shown in Fig. 2.

5

From the global data of SAGE II instrument recorded during the life time of the spacecraft, we filtered out the aerosol extinction data at four channels over Ethiopian region  $(i.e, 3^0N - 15^0N \text{ and } 33^0E - 48^0E)$  for the entire years and it was imported into MATLAB code then processed to calculate the stratospheric aerosol optical depth which is an input to the retrieval algorithm (Mie algorithm) which ultimately provides ASD. Then we calculated the monthly mean aerosol optical depth (AOD) from the measured extinction data by multiplying each extinction value by the level height (i.e, 0.5km) for each channel. Finally, to determine the stratospheric region, the annual mean temperature profile in the same region is plotted to