# Peer review of "Stratospheric Aerosol Climatology over Ethiopia and Retrieval of its Size Distribution"

_Atmospheric Chemistry and Physics, 2017_

## Referee Comment (RC1) · Anonymous Referee #1 · 27 Feb 2017

Reviewer Opinion:

ACPD Paper: acp-2017-133

Title: Stratospheric Aerosol Climatology over Ethiopia and Retrieval of its Size Distribution Author: Milkessa Gebeyehu Homa, Gizaw Mengistu Tsidu, and Derese Tekestebrihan Nega

The research reported in the paper is based on incorrect direct cause effect relation between stratospheric aerosols and human-industrial generated aerosols at the surface in Ethiopia. It has been well established from the first IPCC report (IPCC, 1990) that the main contribution of the aerosols generated at the surface (anthropogenic and industrial) is to the aerosols in the troposphere.

The authors also ignore that the main contribution to the aerosols in the stratosphere during the studied period (October 1984 to September 2005) came from volcanic eruptions. If a climatology of the stratospheric aerosols in any region of the planet wants to be produced it should analyze separated the periods of volcanic activity and the non-volcanic period (called stratospheric aerosols background, because of the low amount of aerosols during this periods).

The paper is plagued of inconsistent and non-sense mixture of stratospheric and tropospheric aerosols citations.

If the authors want to study stratospheric aerosols, they should study first the Assessment of Stratospheric Aerosol Properties Report (SPARC, 2006) and it most recent update (Kremser at al., 2016). There they will find all the main information and scientific conclusions (and also the pending and new scientific questions) about stratospheric aerosols. Among the references in both documents are the leading papers on stratospheric aerosols climatology, size distributions determination, etc.

For the authors to conduct a research on the contribution to the industrial boom in Ethiopia to the increase of tropospheric aerosols they cannot use SAGE II aerosols profiles. Instead, they may use AOD from MODIS instruments measurements (TERRA from 2001 to the present and AQUA from 2002 to the present).

The paper should not be accepted.

References:

IPCC, 1990: Climate Change: The IPCC Scientific Assessment [J. T. Houghton, G. J. Jenkins and J. J. Ephraums (eds.)]. Cambridge University Press, Cambridge, United Kingdom and New York, NY, USA, 212 pp.

Kremser, S., et al., 2016: Stratosphericaerosol—Observations, processes, andimpact on climate, Rev. Geophys., 54,doi:10.1002/2015RG000511.

SPARC, 2006: Assessment of Stratospheric Aerosol Properties (ASAP),WCRP-124,

WMO/TD No. 1295, SPARC Rep. 4, 348 pp.

---

## Referee Comment (RC2) · Anonymous Referee #2 · 15 Mar 2017

Evaluation of ACPD Paper: acp-2017-133

Title: Stratospheric Aerosol Climatology over Ethiopia and Retrieval of its Size Distribution Author: Milkessa Gebeyehu Homa, Gizaw Mengistu Tsidu, and Derese Tekestebrihan Nega

General Comments:

I read all the paper, and I am so confused, there is a mixture of inconsistent issues about stratospheric and tropospheric aerosols during the introduction, analysis, discussion and conclusion in the text. I think that the study bases are wrong. Authors assumed in the study the human industrial generated aerosols are related with the stratospheric aerosols. This is not true, it is widely studied, recognized and stablished that the principal contribution for the stratospheric aerosols are the volcanic eruption

and the surface human generated aerosols contribute to the aerosols concentration in the troposphere. These issues were explained and well stablished since the first ICCP report (ICCP, 1990). Recently, the last year a comprehensive assessment of the stratospheric aerosols was published (Kremser et al., 2016). This report mention "there are evidence of that stratospheric aerosol can also contain small amounts of nonsulfate matter such as black carbon and organics" But also mention that large uncertainties remain with respect to the contribution from anthropogenic sulfur dioxide emissions. So, I think if the objective of the author of the revised paper is to study the stratospheric aerosols above Ethiopia, first they need to read this paper and make use of the results reported there. For example, the main question: How do you demonstrate that human – industrial aerosols produced in Ethiopia influence the stratospheric aerosols? The aerosols from volcanic eruptions mask the influence of other source on the background of stratospheric aerosols, so you need to separate these two periods to study the influence of the human produced aerosols.

The method to study the stratospheric aerosols is not so strict and it is not well explained in the text. There is a mixture in the analysis, between stratosphere and troposphere again, without well explained relation. Authors analyze stratospheric profiles of extinction coefficient and the column AOD (troposphere plus stratosphere), I guess because this is not explained in method section. There is not information about which version of the SAGE II dataset used the authors. Together with these points the method to separate the aerosols and clouds are not analyzed or mentioned. Also, there are a lot of papers and reports studying the stratospheric aerosols with SAGE II, specifically related with SPARC project with an Assessment of Stratospheric Aerosols (ASAP),WCRP-124, 2006). So, the results in the paper in review it is so questionable.

My conclusion and recommendation is the paper should not be accepted for publication.

IPCC, 1990, Climate Change: The IPCC Scientific Assessment. Report prepared for IPCC by Working Group 1. J. T. Houghton, G. J. Jenkins and J. J. Ephraums (eds.).

Cambridge University Press, Cambridge, Great Britain, New York, NY, USA, and Melbourne Australia. 410 pp.

Kremser, S., et al., 2016, Stratospheric aerosol — Observations, processes, and impact on climate, Rev. Geophys., 54, 278–335, doi:10.1002/2015RG000511.

---

## Author Comment (AC1) · 30 Jun 2017

Response to Anonymous Referee

General Response.

The objective of the paper is to report the stratospheric aerosol climatology over Ethiopian sky (in tropics) and characterizing in terms of its physical and optical properties. As the stratosphere level over the region is above 15 km, we have discussed the vertical distribution of aerosols in this region as presented in Figurers 3, 5, 7 and 9. The reason we refer troposphere (usual the upper troposphere) in different parts of the paper is that in our region the tropopause level is a bit higher than 15 km and we considered the region below $\sim$17 km as upper troposphere.

[Figure]

The stratospheric aerosol trend before and after the 1991 (Mt. Pinatubo) volcanic eruption is discussed in terms of its reaction rate at 525 nm, and reveals that the period from 1992 to 2005 is relatively volcanically quiescent period as confirmed by (Thomason L.W. et al 2007)

Response to Anonymous Referee #1

1. Stratospheric aerosols are smaller in size and have longer life time in the atmosphere (Israel H. and Israel, G.H (1974)). Lower stratospheric and upper tropopause regions have almost similar chemical composition (SPARC, 2006). The transport of aerosols formed in the troposphere to stratosphere is mentioned in the review made by Kremser et.al 2016 (page 3 line #9) though the type is not specified.

2. The tropospheric aerosols are not our main focus, but mentioned on the way discussing the stratosphere. It is to mention the lower stratosphere and we will correct it.

3. Sufficient emphasis was given to volcanic eruption as a source of stratospheric aerosols as mentioned on different lines of the article on the discussion paper, mentioned below:

page 2: line 31-33

page 9: line 9-9

page 10: line 9-12

line 16-17

page 11: line 19-21

4. The tropospheric citations were not used directly to stratospheric aerosol discussions. They are used for common aerosol properties undifferentiated.

References

1. Thomason L. et al. (2007). Report on the Assessment of Stratospheric Aerosol Properties: New Data Record, but no Trend.

2. Israel H. and Israel G.W. (1974). Trace Elements in the Atmosphere. Ann Arbor, Michigan.

3. SPARC, 2006: Assessment of Stratospheric Aerosol Properties (ASAP),WCRP-124, WMO/TD No. 1295, SPARC Rep. 4, 348 pp.

4. Kremser, S., et al., 2016: Stratospheric aerosol-Observations, processes, and im-pact on cli-mate, Rev. Geophys., 54,doi:10.1002/2015RG000511.
* * *